# Sonochemical Synthesis of Cu@Pt Bimetallic Nanoparticles

**DOI:** 10.3390/molecules27165281

**Published:** 2022-08-18

**Authors:** Henrik E. Hansen, Daniel Ø. Fakhri, Frode Seland, Svein Sunde, Odne S. Burheim, Bruno G. Pollet

**Affiliations:** 1Electrochemistry Group, Department of Materials Science and Engineering, Faculty of Natural Sciences, Norwegian University of Science and Technology (NTNU), NO-7491 Trondheim, Norway; 2Hydrogen Energy and Sonochemistry Research Group, Department of Energy and Process Engineering, Faculty of Engineering, Norwegian University of Science and Technology (NTNU), NO-7491 Trondheim, Norway; 3Green Hydrogen Lab, Pollet Research Group, Institute for Hydrogen Research (IHR), Université Du Québec à Trois-Rivières (UQTR), 3351 Boulevard des Forges, Trois-Rivières, QC G9A 5H7, Canada

**Keywords:** ultrasound, sonochemistry, electrocatalyst, hydrogen, platinum, copper, core-shell

## Abstract

Reducing the amount of noble metals in catalysts for electrochemical conversion devices is paramount if these devices are to be commercialized. Taking advantage of the high degree of particle property control displayed by the sonochemical method, we set out to synthesize Cu@Pt bimetallic nanocatalysts in an effort to improve the mass activity towards the hydrogen evolution reaction. At least 17 times higher mass activity was found for the carbon supported Cu@Pt bimetallic nanocatalyst (737 mA mg^−1^, *E* = −20 mV) compared to carbon supported Pt nanocatalysts prepared with the same ultrasound conditions (44 mA mg^−1^, *E* = −20 mV). The synthesis was found to proceed with the sonochemical formation of Cu and Cu_2_O nanoparticles with the addition of PtCl_4_ leading to galvanic displacement of the Cu-nanoparticles and the formation of a Pt-shell around the Cu-core.

## 1. Introduction

The sonochemical synthesis method has shown great promise with its ability to easily tune the size of certain noble metal nanoparticles and to achieve narrow nanoparticle size distributions [1,2,3,4,5]. This is achieved through easily adjustable ultrasound parameters such as the ultrasonic frequency and acoustic power which directly control the generation of reducing agents in the form of radicals [4,6,7,8]. Such control over nanoparticle properties is essential when developing new nanomaterials for a range of different applications spanning a broad spectrum of fields including the field catalysis.

To develop better catalyst materials, optimizing the surface properties of the catalyst such as the surface area and the particle size is of great importance [9]. Current wet chemical synthesis methods do not offer such ease of control as particle properties are usually adjusted by precise ratios of surfactants and precursors making reproducibility difficult [9,10]. The sonochemical synthesis method could therefore offer a greater degree of control over these properties thereby paving the way for optimized catalyst materials.

However, many of the best catalyst materials in applications such as water electrolyzers and fuel cells are heavily dependent on noble metals which makes industrial scale production not feasible due to the high cost of these raw materials [11,12]. The hydrogen evolution reaction is one of those reactions in which noble metals such as platinum have been used extensively. Switching to cheaper catalyst materials is a possible solution, but the reduction in price is typically also followed by a reduction in performance. Another possible solution is to minimize the amount of noble metals by synthesizing core-shell nanocatalysts with the noble metal making up the shell and a cheaper metal making up the larger core [13,14]. The noble metal surface properties may therefore be preserved or improved with a reduction in raw material cost.

Several types of core-shell catalysts have been synthesized through various wet-chemical techniques, but one that has received much attention due to its excellent hydrogen evolution and hydrogen oxidation activity is the Cu@Pt bimetallic structure [13,14,15,16]. The general approach to this synthesis appears to follow a two-step process in which the Cu core is synthesized in the first step, followed by galvanic displacement of the metallic Cu by a Pt-precursor to make up the shell. One such synthesis was performed by Zhu et al. [13] where they synthesized Cu@Pt bimetallic nanoparticles with a two-step polyol method using ethylene glycol as the reducing agent. Sarkar et al. [14] also synthesized Cu@Pt bimetallic nanoparticles, where they initially formed metallic Cu through reduction of CuSO_4_·5H_2_O by NaBH_4_ before creating the Pt-shell by galvanic displacement using H_2_PtCl_6_·6H_2_O.

The synthesis of core-shell nanoparticles through a sonochemical route, however, is less common and has mostly been performed with probe sonicators taking advantage of the massive temperature increases from acoustic cavitation [17,18]. Direct sonication with probe sonicators has been shown to cause probe erosion which introduces unpredictable nucleation and growth conditions in turn influencing the particle properties and the purity of the catalyst [19]. Higher frequency plate transducers avoid the problem of erosion all together [19] meaning the high degree of particle control associated with the sonochemical synthesis can be realized. A successful high frequency sonochemical synthesis may therefore offer an easy, reliable way of producing better and cheaper catalyst materials.

In this work we want to determine if the sonochemical synthesis method can be used to synthesize carbon supported Cu@Pt bimetallic nanoparticles, and if the resulting catalyst can challenge a carbon supported Pt-nanocatalyst with respect to its mass activity for hydrogen evolution. Both catalysts were synthesized sonochemically at the same ultrasound conditions to eliminate any effect of the synthesis method. Assessment of the existence of the core-shell structure was performed through physical characterization such as X-ray diffraction and UV-Visible spectroscopy, as well as electrochemical characterization through cyclic voltammetry and linear sweep voltammetry. The hydrogen evolution performance was evaluated based on mass activity at an overpotential of 20 mV.

## 2. Experimental

### 2.1. Sonochemical Setup

For all sonochemical experiments a plate transducer (70 mm Ø) with resonance frequency of 346 kHz from Honda Electronics was used to generate the ultrasonic waves. The piezoelectric material making up the transducer is lead zirconate titanate (PZT), while the plate which transfers the ultrasonic waves to the solution is a highly corrosive resistant stainless steel alloy (SUS304). These plate transducers make up the bottom part of the sonochemical reactor. The rest consists of an inner chamber where the solution is sonicated, and an outer chamber where water is circulating to control the temperature. The top part contains several inlets and outlets to maintain proper atmosphere inside the reactor during sonication. The reactor itself is made from borosilicate glass. A detailed figure showing the different parts of the reactor can be found elsewhere [3]. To generate the ultrasonic waves, an AG 1012 RF signal generator was used in conjunction with an impedance matching unit (T1k-7A) both from T&C Power Conversion. If there is a mismatch between the impedance of the incoming electrical signal from the signal generator and the operating impedance of the piezoelectric transducer, the actual acoustic power will be severely reduced. A mismatch in impedance can be detected as a reflected power with the AG 1012 RF signal generator. By adjusting the impedance matching unit appropriately, near zero reflected power can be achieved, which makes the transducer more efficient.

### 2.2. Synthesis

Sonochemical synthesis of Cu@Pt bimetallic nanoparticles supported on XC-72 carbon was performed at 346 kHz through direct sonication. 2 mmol dm^−3^ CuSO_4_ · 5H_2_O was prepared in a 200 mL 0.2 mol dm^−3^ formate/formic acid buffer of pH 5, and sonicated for 6 h. A pH of 5 was chosen as previous radiolytic investigations of Cu(II) reduction showed that the pH range between 2 and 8 yield metallic Cu [20]. The core-shell structure was achieved through galvanic displacement by adding 50 mL of 0.2 mmol dm^−3^ PtCl_4_ precursor towards the end of the sonication [14]. Samples taken before the addition of PtCl_4_ will from now on be referred to as Cu_2_O-samples, while samples taken after the addition of PtCl_4_ will be referred to as Cu@Pt-samples.

The formate ion is an excellent radical scavenger and a concentration of 0.2 mol dm^−3^ is considered sufficient for radical scavenging without greatly affecting the collapse conditions of the cavitation bubbles [21]. The temperature was maintained at 3 ${}^{{}^{\circ}}C$ with water cooling through the external jacket on the sonochemical reactor. The applied electrical power was kept constant at 50 W, but the effective acoustic power was measured to be 38(3) W. To avoid exposure to O_2_ and N_2_, Ar was supplied to the solution 20 min prior to sonication and throughout the entire synthesis. The resulting particles were extracted from solution using an Eppendorf 5810 R centrifuge operated at 12,000 rpm for 15 min, before being cleaned three times with an ethanol water mixture, and then dried in air over night.

To compare the performance of any Cu@Pt bimetallic catalysts, Pt-nanoparticles supported on XC-72 carbon were also synthesized sonochemically using 0.8 mol dm^−3^ ethanol as a radical scavenger. All other ultrasound parameters were kept the same as described above. These samples will from now on be referred to as Pt-samples.

### 2.3. Characterization

Periodic absorbance measurements were performed throughout the sonication period with an Evolution 220 UV-Visible spectrophotometer. The resulting absorbance spectra were used to assess the development of the metal ions throughout the sonication period. Prior to these measurements, the samples were filtered through a 0.45 μm particle filter from VWR to avoid interference from any precipitates as well as the carbon support.

The resulting Pt-, Cu-, and Cu@Pt-samples were characterized by X-ray diffraction (XRD) in order to determine the crystallographic phases as well as the mean crystallite sizes of these phases. Particles were dispersed in ethanol and drop cast onto flat silicon wafers and dried before being covered by a Kapton film. The measurements were performed with a Bruker D8 A25 DaVinci X-ray Diffractometer with CuKα radiation. A scan rate of 0.044°/step was used for 2*θ*-angles between 30°–75° with a 0.3° fixed slit for 60 min.

Electrochemical characterization was performed in a round bottom electrochemical glass cell with a three electrode configuration. A glassy carbon rotating disk electrode (Pine Instrument) coated with the catalyst ink was used as the working electrode, graphite was used as the counter, and the reversible hydrogen electrode (RHE) was used as the reference electrode. The catalyst ink was prepared by mixing 10 mg catalyst material with 475 μL isopropyl alcohol (technical, VWR), 475 μL ultrapure water (Milli-Q, 18.2 MΩ cm), and 50 μL Nafion 117 (5 wt% in mixture of lower aliphatic alcohols and water, Sigma Aldrich). 10 μL of the catalyst ink was spin coated onto the glassy carbon working electrode with a rotation rate of 200 rpm. As the metal loading on the carbon support was estimated to be 20 wt%, the Pt-loading on the glassy carbon for the Pt-sample amounted to 100 μg cm^−2^. For the Cu@Pt-sample, the Pt-loading was even lower with an estimated value of 7.1 μg cm^−2^ assuming complete reduction of the PtCl_4_ precursor.

All measurements were conducted in an Ar-saturated 0.5 mol dm^−3^ H_2_SO_4_ (95–97%, VWR) electrolyte. Cyclic voltammetry was performed for potentials between 0.04 V–1.6 V vs. RHE with a scan rate of 50 mV s^−1^. From the hydrogen under deposition peaks located between 0.04 V–0.3 V vs. RHE the electrochemical active surface area (ECSA) was estimated using a monolayer coverage value of 220 μC cm^−2^ [22]. The activity towards the hydrogen evolution reaction was also investigated through linear sweep voltammetry with a scan rate of 1 mV s^−1^. Mass activities at −20 mV vs. RHE were compared between the carbon supported Cu@Pt bimetallic nanocatalysts and the carbon supported Pt nanocatalysts.

### 2.4. Transducer Characterization

The ultrasonic plate transducer was subject to rigorous testing before synthesis in an effort to determine its actual acoustic power and radical generation capabilities. To determine the acoustic power, calorimetric measurements were performed in which the temperature increase caused by sonication was measured with the assumption that all acoustic power is transformed into heat [23,24]. Sonication was performed in 200 mL of deionized water at room temperature over the course of 1 min. The acoustic power can therefore be calculated as
(1)$$P_{\mathrm{acoustic}}=mC_{p}{\left(\frac{dT}{dt}\right)}_{t=0}$$
where (*dT*/*dt*)_*t*=0_ is the temperature slope of water per unit of sonication time (at *t* = 0), *m* is the mass of the water, and *C*_p_ is the specific heat capacity of water (4.186 J g^−1^ K^−1^) [23,24].

As for the radical generation capabilities of the ultrasonic transducer, TiOSO_4_ dosimetry was used to measure the yield of H_2_O_2_ generated by sonication [8]. As H_2_O_2_ is the primary recombination product of the ·OH radicals in pure water systems
$$2\begin{array}{c} \mathrm{OH}\;\cdot \; \end{array}⟶\begin{array}{c} H_{2}O_{2} \end{array}$$

We can assume that the yield of ·OH radicals is twice as high as the concentration of H_2_O_2_. The detection of H_2_O_2_ was performed spectrophotometrically by measuring the light absorption of a Ti-complex forming when excess TiOSO_4_ is added to a sonicated sample of water. The TiOSO_4_ reacts with H_2_O_2_ to form this yellow Ti-complex which has a molar absorption coefficient of 787 mol^−1^ dm^3^ cm^−1^ at 411 nm in the absorbance spectrum [3]. Sampling from 0 min to 20 min was performed in order to obtain the rate of ·OH radical formation for each frequency. Longer sonication times were not necessary as the ·OH radical formation is expected to follow zero-order kinetics [3].

## 3. Results

Absorbance spectra for the Cu(II) species are shown in Figure 1a for different sonication times. The decrease in absorbance with sonication time indicates that the Cu(II) concentration also decreases during sonication signifying a successful reduction by the sonochemically generated radicals. Plotting the absorbance at the peak (773 nm) as a function of sonication time (Figure 1b) shows that the reduction rate follows first-order kinetics with a rate constant of *k* = 0.27 h^−1^. Such a first-order behaviour is similar to the previously observed sonochemical reduction of Pt(IV) to Pt(II), albeit with a much lower rate constant [3].

After sonicating the Cu-solution for 6 h, PtCl_4_ was added to the sonicating solution and the development of the Pt(IV) concentration was measured with UV-Visible spectroscopy for 10 min. To enhance the Pt(IV) peaks, the sample was mixed with an excess of KI to form PtI_6_^2−^ ($\lambda_{\max}$ = 495 nm) [25]. The resulting absorbance spectra are shown in Figure 2a. The figure shows a clear decrease in the peak at 495 nm suggesting the successful reduction of Pt(IV) to Pt(II) as indicated by the appearance of a PtI_4_^2−^ peak at 388 nm [25].

The absorbance belonging to the Pt(IV) species at 495 nm is also plotted as a function of sonication time in Figure 2b. The Pt(IV) concentration is shown to decrease rapidly to about 50% of the starting concentration within the first minute before it experiences a slow increase. Further sonication appears to stabilize the Pt(IV) concentration close to the value obtained after 1 min.

X-ray diffractograms of the resulting particles from the sonochemical synthesis are shown in Figure 3. The Cu@Pt-sample (c) exhibits peaks corresponding to metallic Cu at 2θ angles of 43.3° (111), 50.4° (200), and 74.1° (220). Additional peaks corresponding to Cu_2_O are also found at 2θ angles of 29.5° (110), 36.4° (111), 42.3° (200), 61.4° (220), and 73.4° (311). No clear indication of Pt was found in the diffractogram, but a separate diffractogram of Pt supported on carbon (a) shows that its contribution disappears into the carbon support background. The Cu_2_O-sample (b) only exhibits peaks corresponding to Cu_2_O as described above.

Applying the Scherrer equation to the Cu_2_O (111) and metallic Cu (111) peaks showed that the crystallite size for the Cu_2_O-sample was significantly larger (84.1 nm) than the Cu@Pt-sample (35.0 nm). No determination of the crystallite size for the Pt-sample was feasible due to its signal being quenched by the carbon support.

The cyclic voltammograms of the Pt-, Cu_2_O-, and Cu@Pt-samples are shown in Figure 4. For the Pt-sample (a), the characteristic CV of Pt was observed [13] confirming that Pt was indeed prepared sonochemically. From the hydrogen underpotential deposition peaks, the ECSA was estimated to be 6.2 m^2^ g^−1^. For the Cu_2_O-sample (b), no clear features are observed in the voltammogram at lower potentials. If the particles contained metallic Cu, one would expect to see Cu dissolution represented by a stripping peak around *E* = 0.6 V for the first cycle in the anodic direction [26]. Towards higher potentials, the Cu_2_O-sample displays an increased current, even surpassing that of the Pt-sample. For the Cu@Pt-sample (c), the characteristic voltammogram of Pt can be seen [13], albeit with a smaller current compared to the Pt-sample. This can most likely be attributed to the lower Pt-loading as expected for the Cu@Pt-sample. A voltammogram acquired at 400 mV s^−1^ (Appendix A) was used to estimate the ECSA of the Cu@Pt-sample in order to enhance the hydrogen underpotential deposition peaks. The resulting ECSA was estimated to be 15.1 m^2^ g^−1^.

The catalytic activity of the Pt-, Cu_2_O-, and Cu@Pt-samples towards the hydrogen evolution reaction (HER) are shown through linear sweep voltammetry (Figure 5). The Pt-sample displays an overpotential of 29.3 mV at 10 mA/cm^2^_geo_ towards the HER, while the Cu_2_O-sample has zero contribution to the HER in the same region. The Cu@Pt-sample has a very similar performance to the Pt-sample with an overpotential of 28.6 mV at 10 mA/cm^2^_geo_ towards the HER. The mass activity of the Pt-sample at an overpotential of 20 mV was found to be 44 mA mg^−1^ while the mass activity of the Cu@Pt-sample was found to be 737 mA mg^−1^. This amounts to an approximately 17 times higher mass activity for the Cu@Pt-sample.

Results from dosimetry and calorimetry experiments are shown in Figure 6a,b, respectively. The TiOSO_4_ dosimetry shows an approximately linear increase in the ·OH radical formation. This is also expected as zero-order kinetics has been demonstrated for a similar ultrasound configuration previously [3]. The radical generation rate is estimated to be (12 μmol dm^−3^ min^−1^) assuming all ·OH radicals recombine into H_2_O_2_. The corresponding absorbance spectra are given in the Appendix A. From the calorimetry measurements the acoustic power was estimated using Equation (Equation 1) to be 38(3) W. The temperature increase in the reactor as a function of sonication time is given in Figure 6b.

## 4. Discussion

The sonochemical synthesis of Cu@Pt bimetallic nanoparticles appears to be possible by taking advantage of galvanic displacement of Cu by Pt(IV). The characteristic CV of Pt and the enhanced mass activity observed through electrochemical measurements (Figure 4 and Figure 5b, respectively) as well as the existence of metallic Cu as seen from XRD (Figure 3) show that the sample contains both Pt and Cu.

If Pt and Cu were to be separate phases, any metallic Cu would oxidize to Cu_2_O upon exposure to air during drying as was observed for the Cu_2_O-sample (Figure 3). In addition, we would have seen an increase in the current towards higher potentials in the cyclic voltammogram as was demonstrated by the Cu_2_O-sample (Figure 4). The much higher mass activity observed for the Cu@Pt-sample (Figure 5b) is also an indication of a core-shell structure as such low Pt-loadings would not be able to reach such high currents. This was demonstrated by the Pt-sample which reached the same current densities as the Cu@Pt-sample (normalized for geometric surface area) at a much higher Pt-loading (Figure 5a). One can also argue that the presence of Pt(IV)- and Pt(II)-ions in the reaction solution (Figure 2a) after the reaction was completed is an indication of the Cu-sites being fully blocked by a shell of Pt resulting in no further Pt-reduction.

Arguments for the galvanic displacement of Cu by Pt(IV) can be made through careful analysis of the UV-Vis absorbance spectra of Pt (Figure 2a) and the radical generation capabilities of the sonochemical reactor as measured through TiOSO_4_ dosimetry (Figure 6a). From the Pt UV-Vis absorbance spectra in Figure 2a, the rapid reduction of Pt(IV) to about 50 % of the starting concentration within the course of 1 min is a clear indication of an assisted reduction. The ·OH radical generation rate was determined to be 12 μmol dm^−3^ min^−1^ as shown through dosimetry measurements (Figure 6a). If all ·OH radicals generated within this minute was fully scavenged by the formate ions, it would be able to reduce maximum 1.5 μmol Pt(IV) to Pt(II), and only 0.75 μmol Pt(IV) to Pt(0). The amount of Pt(IV) being added to the reactor is 10 μmol which means that if half of this is reduced within the first minute, the sonochemical synthesis must be assisted. With Pt exhibiting a more positive reduction potential than Cu, the galvanic displacement of Cu with Pt(II) and Pt(0) therefore appears to be the dominant reduction mechanism for the added Pt(IV).

The estimated crystallite sizes of the Cu-phases from XRD for the Cu_2_O-sample (84.1 nm) and the Cu@Pt-sample (35.0 nm) also suggests that the size determining Cu-core has been reduced in size upon adding PtCl_4_. Such a size reduction would only occur if some of the already formed Cu is galvanically replaced by Pt or Pt(II). Another complementary effect to a size reduction of the Cu@Pt-sample is a size increase for the Cu_2_O-sample which would occur upon formation or growth of Cu_2_O when the particles are dried in air. Both explanations support the formation of a Cu@Pt bimetallic structure.

The Pt(IV) concentration was also observed to stabilize after the reaction was completed (Figure 2) which could suggest that there are no more Cu-sites available for galvanic displacement. From the X-ray diffractograms (Figure 3) it is shown that metallic Cu still prevails in the sample. The metallic Cu present in the sample must therefore be protected from the Pt(IV) still present in the reactor. This could be due to the formation of a metallic Pt shell around the Cu-particles.

Another argument which can be made for the formation of the Cu@Pt bimetallic structure is the fact that the metallic Cu phase is not present in the X-ray diffractogram of the Cu_2_O-sample taken before the addition of PtCl_4_ as seen in Figure 3. The absence of a covering Pt-shell could allow for the oxidation of metallic Cu to Cu_2_O when dried in air, but if the particles are coated with a layer of Pt no such oxidation is possible. Peaks belonging to Cu_2_O are still seen in the X-ray diffractogram of the Cu@Pt-samples which might suggest that not all particles are perfectly covered with Pt or that this Cu_2_O phase was also formed to some degree during sonication. As was previously discussed, no free Cu sites should be available as this would have ensured further Pt(IV) reduction. Cu_2_O may form in situ through the reaction between metallic Cu and hydrogen peroxide which is an expected byproduct of unreacted ·OH during sonication.
(2)$$2\mathrm{Cu}+H_{2}O_{2}⟶{Cu}_{2}O+H_{2}O$$

This reaction is evidently not dominating as metallic Cu prevails throughout the entire 6 h sonication period, but it can explain why Cu_2_O is observed to some degree for the Cu@Pt-sample. The presence of hydrogen peroxide is further strengthened by the slight increase in Pt(IV) concentration after the initial reduction as seen in Figure 2b. When the Pt(IV) is reduced to Pt(II), Pt(II) may be reoxidized by any residual hydrogen peroxide in the reactor. The hydrogen peroxide is then removed through this reaction, and the Pt(IV) concentration eventually stabilizes. The Cu_2_O phase observed along with the Cu@Pt bimetallic structure therefore appears to be a byproduct of unreacted ·OH radicals during sonication.

## 5. Conclusions

The sonochemical synthesis of carbon supported Cu@Pt bimetallic nanoparticles proved successful with the catalysts achieving at least 17 times higher mass activity compared to carbon supported Pt-nanoparticles prepared in the same manner. The core-shell formation appears to occur through three steps; Cu(II) is reduced to metallic Cu through sonochemical reduction. A Cu_2_O phase then appears to some degree due to oxidation of metallic Cu by hydrogen peroxide. When PtCl_4_ is added, the Pt is galvanically reduced at the metallic Cu surface until all Cu sites are covered with Pt making up the Cu@Pt bimetallic structure.

## Figures and Tables

**Figure 1 molecules-27-05281-f001:**
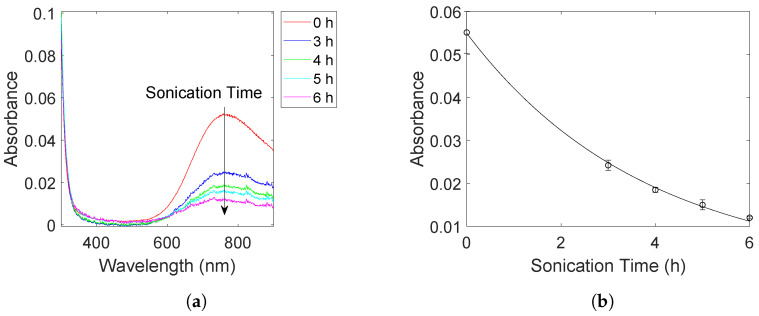
Absorbance spectra of Cu(II) at different sonication times (**a**), and the absorbance at 773 nm plotted as a function of sonication time (**b**). The contribution from other interfering species such as the carbon support and the as-formed nanoparticles were removed by filtration.

**Figure 2 molecules-27-05281-f002:**
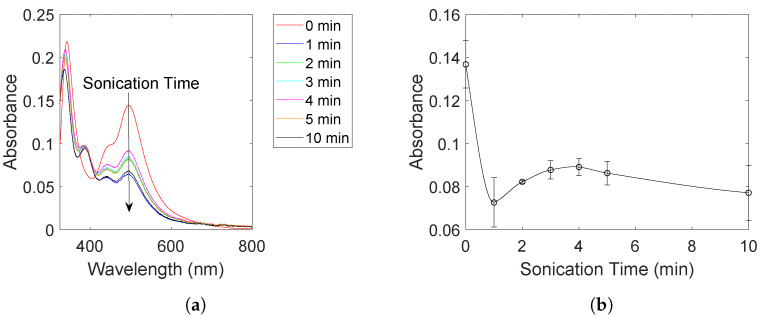
Absorbance spectra of PtI_6_^2−^ and PtI_4_^2−^ at different sonication times (**a**) and the absorbance at 495 nm plotted as a function of sonication time (**b**). The sonication time is measured in minutes after the addition of PtCl_4_ to the Cu-solution. The contribution from other interfering species such as the carbon support and the as-formed nanoparticles were removed by filtration.

**Figure 3 molecules-27-05281-f003:**
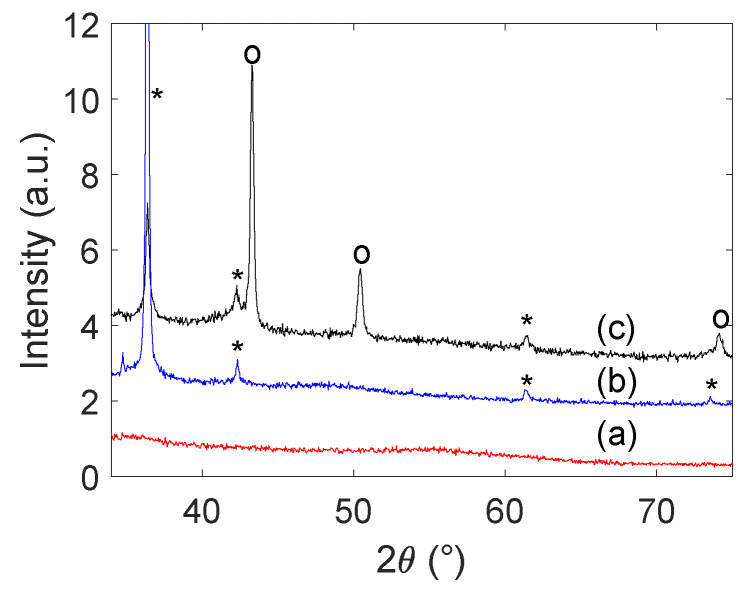
X-ray diffractograms of particles from the sonochemical synthesis. Carbon supported Pt-nanoparticles are shown in red (a), while Cu-samples synthesized before the addition of PtCl_4_ are shown in blue (b), and Cu-samples synthesized after the addition of PtCl_4_ are shown in black (c). Peaks corresponding to the Cu_2_O phase (*) and the metallic Cu phase (o) are marked in the figure.

**Figure 4 molecules-27-05281-f004:**
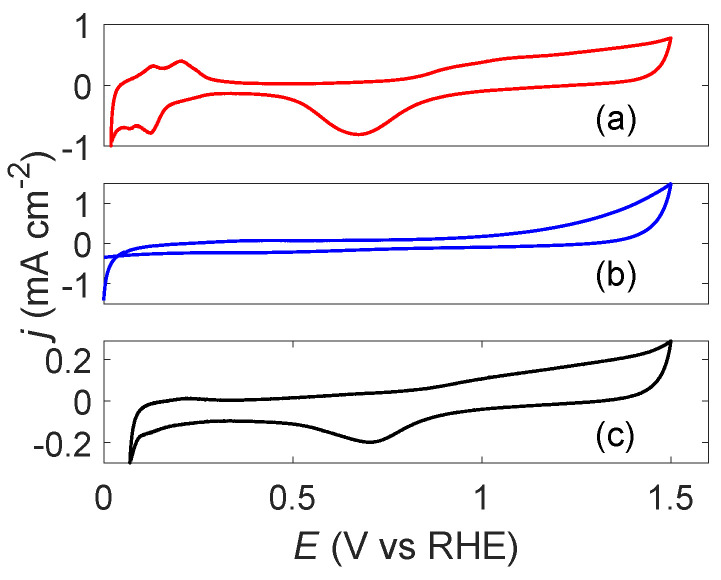
Cyclic voltammograms for sonochemically synthesized carbon supported Pt-nanoparticles (**a**), and Cu-nanoparticles before (**b**) and after (**c**) the addition of PtCl_4_. All measurements were performed in 0.5 mol dm^−3^ H_2_SO_4_ with a scan rate of 50 mV s^−1^.

**Figure 5 molecules-27-05281-f005:**
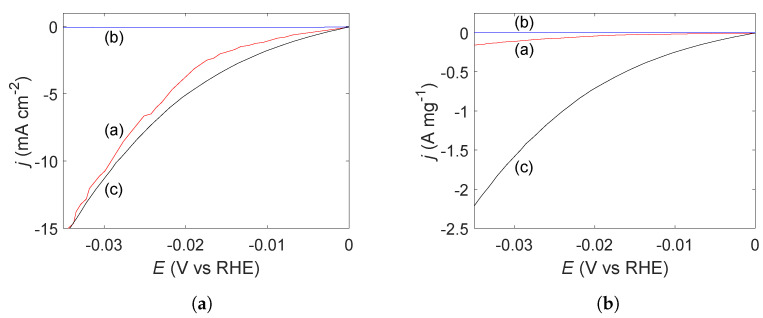
Linear sweep voltammogram acquired over the hydrogen evolution region for sonochemically synthesized carbon supported Pt-nanoparticles (a), and Cu-nanoparticles before (b) and after (c) the addition of PtCl_4_. Figure (**a**) shows the current normalized for the geometric surface area, while Figure (**b**) shows the current normalized for the estimated mass of Pt.

**Figure 6 molecules-27-05281-f006:**
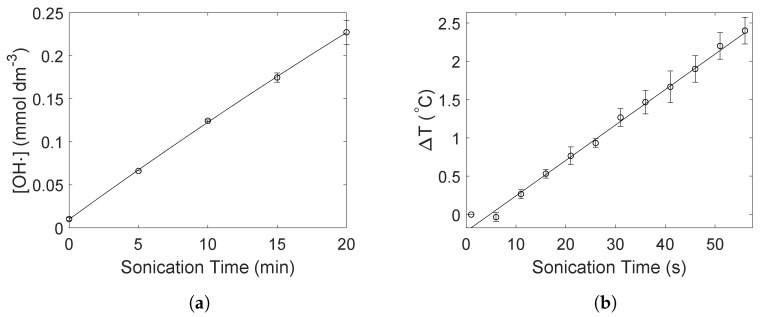
TiOSO_4_ dosimetry showing the ·OH concentration as a function of sonication time (**a**). The concentration of the ·OH was estimated from the hydrogen peroxide concentration. Calorimetry measurements of the acoustic power of ultrasound showing the resulting temperature increase in water as a function of sonication time (**b**).

## Data Availability

Data supporting the results can be obtained by contacting the main author.

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
