# Peer review of "Sonochemical Synthesis of Cu@Pt Bimetallic Nanoparticles"

_molecules, 2022, doi:10.3390/molecules27165281_

Round 1
Reviewer 1 Report
Presented manuscript is devoted to the actual task of the modern ecologically friendly energy storage and generation namely the design of the new nanostructured materials for the development of the electrode for electrochemical hydrogen generation. The presented results are of some interest from a practical point of view. However, I cannot recommend this manuscript for the publication due to lack of characterization of the obtained products. For example, 1) the authors declare about obtaining of the copper nanoparticles but XRD pattern have shown the Cu2O; 2) the structure and the composition of Cu-Pt particles is not investigated. As about the second note, the authors have concluded about the formation of core-shell nanoparticles based on the fact of the presence of Cu(0) on XRD patterns and the inhibition of particles oxidation. But any other evidences of the presence of Pt were not presented. That is why in future the authors should paid more attention to the characterization of the obtained products.
Author Response
The authors would like to thank the reviewer for their comments and helpful suggestions for future work. We have decided to address the reviewer's comments by dividing their answer up into two parts. The reviewer’s comments are shown in italic with the authors reply written below in normal font.
1) The authors declare about obtaining of the copper nanoparticles but XRD pattern have shown the Cu2O.
We thank the reviewer for their comment. Yes, in Figure 3 graph b, only Cu2O is observed. We realize that the naming of this sample as "the Cu sample" is somewhat misleading. We have therefore decided to change any reference of "the Cu-sample" to "the Cu2O-sample" in the hopes that this makes it clearer. As for graph c in Figure 3, the presence of metallic Cu is evident by the additional peaks that have appeared in the diffractogram. This is also stated in section 3 above Figure 3. Cu2O is also present here along with the metallic Cu. The reason for this was discussed in section 4 paragraph 6 and 7.
2) The structure and the composition of Cu-Pt particles is not investigated. As about the second note, the authors have concluded about the formation of core-shell nanoparticles based on the fact of the presence of Cu(0) on XRD patterns and the inhibition of particles oxidation. But any other evidences of the presence of Pt were not presented. That is why in future the authors should paid more attention to the characterization of the obtained products.
We thank the reviewer for their comment. The aim of this work was mainly to show that the sonochemical synthesis method can be used to obtain core-shell nanoparticles and how its catalytic activity compares to Pt alone. Detailed characterization of the composition and structure beyond the existence of core-shell particles therefore falls outside the scope of this paper. As for the existence of Pt the authors agree with the reviewer that there is no evidence of bulk Pt in the catalysts. However, there is clear evidence of Pt in the catalysts as shown by cyclic voltammetry in Figure 4 graph a and c. Both these voltammograms display the characteristics of Pt, and to our understanding and the best of our knowledge, the CVs undeniably document the presence of Pt in a sulfuric acid electrolyte. To make these characteristics more visible, we have decided to change Figure 4. One is then left with three options for how this Pt is distributed in the sample: 1) The Pt is present as individual particles. 2) Pt forms an alloy with the Cu. 3) Pt is present in the form of a protective shell surrounding a core of Cu. We observe elemental Cu in what we have named the “Cu@Pt-sample”, and this rules out that bare Cu can be found in the sample as it would be oxidized to Cu2O as demonstrated by the Cu2O-sample. This rules out option 1 and 2 as both these will have exposed Cu-surfaces. The Cu therefore has to be covered by something as pointed out in the second paragraph of the Discussion in the manuscript already. The only available element is Pt, for which we have evidence is being reduced.
As for the lack of characterization of the resulting particles, the authors agree with the reviewer that more characterization through for example SEM/TEM and EDX would have been great, but we were not able to achieve a suitable resolution with any of our instruments to say anything about the particle structure. Such characterization methods are also quite dependent on sample preparation and are highly qualitative and might not represent the entire sample. Our use of purely quantitative characterization techniques therefore assures us that these properties are representative for the entire sample. It would have been great to have some complementary SEM/TEM images to further back up our results, but this was unfortunately not possible.
Reviewer 2 Report
The manuscript submitted by H.E. Hansen and coworkers reports on the sonochemical preparation of nanocatalysts for hydrogen evolution, composed of copper core and platinum shell. The topic is actual and certainly worth investigating. I recommend publishing the paper, after some minor corrections and changes:
· The English language use in the manuscript is good, with some minor problems. One that I noticed is the incorrect usage of plural verbs, e.g.: (Introduction, 2nd Paragraph): … methods does not offer … should read: methods do not offer. Introduction, 3rd Paragraph: … metals such as platinum has … should read: metals such as platinum have. Discussion, Paragraph 6: Peaks belonging to Cu2S is still seen … should be: Peaks belonging to Cu2S are still seen
· Abstract, line 6: the authors write that they compared the mass activity of Cu@Pt nanocatalysts to Pt nanocatalysts; from the following text it is clear that they compared the activity of Cu@Pt nanocatalysts to Pt-on-carbon nanocatalysts. I suggest that ‘Pt nanocatalysts’ is replaced by ‘carbon supported Pt nanocatalysts’.
· Keywords: they appear to be missing entirely?
· Introduction, Paragraph 1: the authors claim how ‘The sonochemical method has shown great promise in recent years …’, however, at least two of the cited papers (1-5) are anything else then recent ((1) = 1996 and (2) = 2005). More recent references should be used.
· Figure 2a: the yellow (?) line for absorbance after 5 min is almost invisible; please use another color.
· Section 2.3, Paragraph 5 (below Figure 2): The authors write that ‘X-ray diffractograms of the resulting carbon supported particles (…) are shown in Figure 3.’ The same is written in the Caption below Figure 3. However, it is evident that Fig. 3 shows only one type of carbon supported particles, i.e. (a). Particles (b) and (c) are not carbon supported. I suggest removing ‘carbon supported’ from the text and the caption.
· Section 2.3, Paragraph 7 (below Figure 3): I don’t believe that is ‘synthesized’ the correct term for the preparation of Pt from PtCl4 precursor. Synthesis means preparation of new compounds from elements or other compounds, however in this case, a compound (PtCl4) was decomposed to obtain Pt. I suggest ‘Pt was prepared sonochemically’ or ‘Pt was obtained sonochemically’.
However, I also have one major issue with the manuscript: the authors failed to undoubtedly confirm the presence of elemental Pt on their nanoparticles. No peaks of Pt can be seen on the XRD diffractograms (Fig. 3) while absorbance spectra (Fig. 2) only confirm that the concentration of Pt(IV) decreases while those of Pt(II) increases. However, tis is not the proof of the presence of Pt on the nanoparticles. It is true that the existence of metal Cu and the increase of mass activity are good indications of the proposed Cu@Pt structure. Nevertheless, I strongly suggest that the authors perform additional SEM and/or TEM measurements, combined with EDXS, to confirm (a) the core – shell structure and (b) the presence of elemental Pt.
Author Response
We thank the reviewer for their helpful comments and for the thorough assessment of our manuscript. The following answers to the reviewer's comments will be presented with a number followed by the reviewer comment in italic, and our answer in a normal font below.
1) The English language use in the manuscript is good, with some minor problems. One that I noticed is the incorrect usage of plural verbs, e.g.: (Introduction, 2nd Paragraph): … methods does not offer … should read: methods do not offer. Introduction, 3rd Paragraph: … metals such as platinum has … should read: metals such as platinum have. Discussion, Paragraph 6: Peaks belonging to Cu2S is still seen … should be: Peaks belonging to Cu2S are still seen
We thank the reviewer for going through the grammar and helping us achieve a better manuscript. All grammar mistakes which were pointed out by the reviewer has been corrected. We also carefully went through the manuscript and changed a few additional errors. These are highlighted in yellow along with all other changes in the manuscript.
2) Abstract, line 6: the authors write that they compared the mass activity of Cu@Pt nanocatalysts to Pt nanocatalysts; from the following text it is clear that they compared the activity of Cu@Pt nanocatalysts to Pt-on-carbon nanocatalysts. I suggest that ‘Pt nanocatalysts’ is replaced by ‘carbon supported Pt nanocatalysts’.
We thank the reviewer for the suggested paraphrasing. The suggested change from Pt nanocatalysts to carbon supported Pt nanocatalysts has been made.
3) Keywords: they appear to be missing entirely?
We thank the reviewer for pointing out the missing keywords. They have now been implemented below the abstract. We also noticed that the acknowledgements were somehow missing in the manuscript and we have added this as well.
4) Introduction, Paragraph 1: the authors claim how ‘The sonochemical method has shown great promise in recent years …’, however, at least two of the cited papers (1-5) are anything else then recent ((1) = 1996 and (2) = 2005). More recent references should be used.
We thank the reviewer for pointing out this incorrect use of the term "recent years". We have now removed this from the manuscript.
5) Figure 2a: the yellow (?) line for absorbance after 5 min is almost invisible; please use another color.
We thank the reviewer for this suggestion. The yellow color in Figure 2a has been changed to orange. The arrow in the figure was meant to show the trend in the development of the absorbance for Pt(IV), but we agree with the reviewer that all datapoints should be clearly visible.
6) Section 2.3, Paragraph 5 (below Figure 2): The authors write that ‘X-ray diffractograms of the resulting carbon supported particles (…) are shown in Figure 3.’ The same is written in the Caption below Figure 3. However, it is evident that Fig. 3 shows only one type of carbon supported particles, i.e. (a). Particles (b) and (c) are not carbon supported. I suggest removing ‘carbon supported’ from the text and the caption.
We thank the reviewer for the comment, and for discovering this oversight from the authors. We have now removed “carbon supported” from the text and the caption surrounding the XRD-results. After going through the raw data we also discovered that a background subtraction was performed on graph b and c. This has now been corrected so that the background is present in all samples.
7) Section 2.3, Paragraph 7 (below Figure 3): I don’t believe that is ‘synthesized’ the correct term for the preparation of Pt from PtCl4 precursor. Synthesis means preparation of new compounds from elements or other compounds, however in this case, a compound (PtCl4) was decomposed to obtain Pt. I suggest ‘Pt was prepared sonochemically’ or ‘Pt was obtained sonochemically’.
We thank the reviewer for this suggestion, and we have decided to implement this change. It now states that "Pt was prepared sonochemically" in Section 2.3, Paragraph 7.
8) However, I also have one major issue with the manuscript: the authors failed to undoubtedly confirm the presence of elemental Pt on their nanoparticles. No peaks of Pt can be seen on the XRD diffractograms (Fig. 3) while absorbance spectra (Fig. 2) only confirm that the concentration of Pt(IV) decreases while those of Pt(II) increases. However, tis is not the proof of the presence of Pt on the nanoparticles. It is true that the existence of metal Cu and the increase of mass activity are good indications of the proposed Cu@Pt structure. Nevertheless, I strongly suggest that the authors perform additional SEM and/or TEM measurements, combined with EDXS, to confirm (a) the core – shell structure and (b) the presence of elemental Pt.
We thank the reviewer for commenting on the characterization of Pt in the final Cu@Pt core-shell nanocatalysts. For the XRD-diffractograms it was shown through sample a in Figure 3 that Pt supported on carbon will not produce a clearly visible peak as it is overshadowed by the carbon. It will therefore not necessarily be visible in the XRD diffractogram of the Cu@Pt core-shell sample (sample c in Figure 3) either as the Pt-loading is much smaller here, and from the UV-Vis data we also see that not all PtCl4 is reduced which means that we are dealing with a very low amount of Pt compared to Cu. We know that the Pt-sample contains Pt nanoparticles as this synthesis method is based on our previous work (reference [3]) which was cited in the 1st paragraph of section 3. More importantly, the characteristic CV of Pt in Figure 4 (sample a and c) is irrefutable evidence of Pt being present at the particle surface in both the Pt-sample and the Cu@Pt core-shell sample. This is argued briefly in the first paragraph of section 4. No extensive discussion regarding the presence of Pt was conducted as the authors believe the electrochemical measurements (especially the CVs in Figure 4) speaks for themselves. However, we have decided to improve upon Figure 4 by making these characteristics more easily visible. As for the formation of the core-shell structure one is then left with three options for how this Pt is distributed in the sample: 1) The Pt is present as individual particles. 2) Pt forms an alloy with the Cu. 3) Pt is present in the form of a protective shell surrounding a core of Cu. We observe elemental Cu in what we have named the “Cu@Pt-sample”, and this rules out that bare Cu can be found in the sample as it would be oxidized to Cu2O as demonstrated by the Cu2O-sample. This rules out option 1 and 2 as both these will have exposed Cu-surfaces. The Cu therefore has to be covered by something as pointed out in the second paragraph of the Discussion in the manuscript already. The only available element is Pt, for which we have evidence is being reduced.
As for electron microscopy evidence of the core-shell structure, we did attempt SEM characterization, but we were not able to get proper resolution with the EDX to confidently confirm or deny a core-shell structure. However, the authors believe that the current discussion shows that a core-shell structure is reasonably justified.
Round 2
Reviewer 1 Report
I understood the arguments of the authors of the manuscript regarding the structure of nanoparticles and I consider them quite logical. However, at this manuscript it should be better to mark the obtained product as "Cu@Pt bimetallic nanoparticles", no "core-shell". At least until clear and obvious evidence of the formation of the "core-shell" structures will be obtained.
Author Response
We thank the reviewer for the comment, and have decided to implement the proposed change in the naming of the obtained product. The obtained product is therefore now changed from "Cu@Pt core-shell nanoparticles" to "Cu@Pt bimetallic nanoparticles" as per the reviewer's suggestion.
Reviewer 2 Report
The authors have now corrected all issues and provided good answers to the comments. I believe the paper is now sufficiently improved to recommend it to be published in its present form.
Author Response
We thank the reviewer for accepting the changes and for helping to improve the manuscript.